# THE GEOMETRY OF SELF-VERIFICATION IN A TASK-SPECIFIC REASONING MODEL

## ABSTRACT

How do reasoning models verify their own answers? We study this question by training a model using DeepSeek R1's recipe on the CountDown task. We leverage the fact that preference tuning leads to mode collapse, yielding a model that always produces highly structured chain-of-thought sequences. With this setup, we do top-down and bottom-up analyses to reverse-engineer how the model verifies its outputs. Top-down, we find Gated Linear Unit (GLU) weights encoding verification-related tokens, such as "success" or "incorrect". Bottom-up, we find that "previous-token heads" are mainly responsible for self-verification in our setup. Our analyses meet in the middle: drawing inspiration from inter-layer communication channels, we use the identified GLU weights to localize as few as six attention heads that can disable self-verification, pointing to a necessary component of a potentially larger verification circuit. Finally, we verify that similar verification components exist in our base model and a general reasoning DeepSeek-R1 model.

## 1  INTRODUCTION

Recent language models demonstrate impressive reasoning capabilities. Such models are trained with reinforcement learning (RL) in which they are rewarded when their final outputs are correct.

Behaviorally, these models generate long chain-of-thought (CoT) (Wei et al., 2022) reasoning traces. There is an open question on whether monitoring their CoT is worthwhile, given a growing line of work suggesting that their CoTs do not faithfully reflect the model's inner computations (Turpin et al., 2023; Lanham et al., 2023; Arcuschin et al., 2025). Can we monitor their hidden states instead? We take a step towards investigating this question, by studying a model's inner mechanism for a crucial reasoning step, i.e., self-verification.

General reasoning entails a broad range of tasks, requiring a diverse set of skills. In order to conduct a systematic study, we train and analyze a task-specific reasoning model using the same recipes from DeepSeek R1. We limit the scope of our study to a specific task that requires search – a core reasoning skill broadly applicable for many reasoning tasks. We also select a task in which we can expect the verification mechanism ahead of time, making our analyses easier. Namely, we study CountDown Gandhi et al. (2024; 2025); Qin et al. (2025); Yao et al. (2023), in which a set of numbers (operands) and a target number is given, and the model must find the right arithmetic combination using the operands to reach the target number. Because the target number is specified in the context, we can expect attention heads to play a role in verification and shed light onto other relevant weights and subspaces pertaining to self-verification.

Studying a task-specific model has a second non-obvious benefit: training language models with RL (i.e., with preference signals) can lead to mode collapse towards majority preferences, significantly reducing the diversity of their outputs (Kirk et al., 2024; Padmakumar & He, 2024; Murthy et al., 2024; Slocum et al., 2025). Luckily, in the context of model interpretability, this means that our task-specific model converges to always generating well-structured CoT sequences, allowing us to easily and systematically parse its reasoning trace (e.g., see Table 1).

We conduct "top-down" and "bottom-up" analyses to reverse-engineer how the model verifies its own outputs. Our two analyses meet in the middle, revealing key subspaces relevant for model verification.

Going top-down, we leverage linear probes to find Gated Linear Unit (GLU) vectors in late layers that often encode tokens relevant for verification. Interestingly, these vectors also seem to correlate with

English or Chinese tokens, like "success" or "不完" ("failed"). Furthermore, the antipodal directions of these vectors also encode the antonyms of correct or incorrect tokens.

Going bottom-up, given the nature of our task, we hypothesize and verify that attention heads play a significant role. We find "previous-token heads" – attention heads that attend to previous occurrences of the current token – that attend to the provided solution in the context. Previous-token heads have been studied before, for instance in the context of induction heads (Olsson et al., 2022). Through causal analyses, we find that disabling previous-token heads disables model verification.

Our two analyses meet in the middle: we find that disabling previous-token heads also deactivates our GLU vectors. Inspired by inter-layer communication channels (Elhage et al., 2021; Merullo et al., 2024), we look for previous-token heads that most align with the "receptive-field" of our GLU vectors, allowing us to localize as few as six attention heads that reliably disables model verification. Thus our work finds necessary components for a potentially larger verification circuit.

Finally, we verify that similar verification components exist in our base model prior to RL, as well as in a general reasoning model, `DeepSeek-R1-Distill-Qwen-14B`.

Obviously, most reasoning tasks do not provide an easily verifiable solution in the context. However, by illustrating a thorough mechanism of verification in our simplified setup, we take a step towards the possibility of monitoring and interpreting a model's inner computations in its hidden states.

## 2 NOTATIONS, KEY TERMINOLOGIES

We first establish key terminologies and notations. A Transformer's forward pass first embeds the input using weights $W_E \in \mathbb{R}^{d \times V}$. The embeddings go through $L$ Transformer blocks, yielding hidden states $\mathbf{x}^\ell \in \mathbb{R}^d, \ell \in [L-1]$. The last layer, $\mathbf{x}^{L-1}$, is "unembedded", or projected back to the token embedding space using $W_E$, and the nearest neighboring token embedding of $W_E^\top \mathbf{x}^{L-1}$ is outputted. Each block consists of attention heads and Gated Linear Units (GLUs) (Shazeer, 2020).

**Attention.** Each attention head consists of key ($W_K$), query ($W_Q$), value ($W_V$), and output ($W_O$) weights. an attention pattern $A$ is computed using key and query weights:

$$A = \text{softmax}(\mathbf{x}_i^\top W_Q^\top W_K \mathbf{x}_j) \tag{1}$$

where $W_Q^\top W_K$ is sometimes referred to as a "QK circuit". $A$ is used to scale the "OV circuit" ($W_O W_V$) to produce an output for each head:

$$h(\mathbf{x}) = (A \otimes W_O W_V) \cdot \mathbf{x} \tag{2}$$

**Gated Linear Units and GLU$_{\text{Out}}$ Vectors.** Given a Gated Linear Unit (GLU) block:

$$\text{GLU}(\mathbf{x}) = (\phi(W_{gate}\mathbf{x}) \odot W_{up}\mathbf{x}) W_{out} \tag{3}$$

where $\phi$ is a non-linearity and $W_{gate}, W_{up}, W_{out} \in \mathbb{R}^{d_{glu} \times d}$, we decompose it as following:

$$M = \phi(W_{gate}\mathbf{x}) \odot W_{up}\mathbf{x}, M \in \mathbb{R}^{d_{glu}}, \tag{4}$$

$$\text{GLU}(\mathbf{x}) = M^\top W_{out} = \sum_j^{d_{glu}} m_j \mathbf{v}_j \tag{5}$$

where $m_j$ is the $j$-th entry in $M$ and $\mathbf{v}_j$ is the $j$-th row of $W_{out}$.

Importantly, the output of each GLU block can be thought of as the weighted sum of $\mathbf{v}$'s (the rows of $W_{out}$), where the weights are determined by activations $M$. **We refer to elements of $M$ as GLU neurons and vectors $\mathbf{v}$ as GLU$_{\text{Out}}$ vectors**.

## 3 TRAINING TASK-SPECIFIC REASONING MODELS

We apply DeepSeek R1-Zero's setup with Qwen2.5-3B as our base model (Hyperparams: Appx. A).[1]

---

[1]We use TinyZero: `https://github.com/Jiayi-Pan/TinyZero/tree/main`

Table 1: **CountDown Task.** The model must find an arithmetic combination of the operands to reach the specified target number. We leverage the fact that preference tuning leads to mode collapse, resulting in a model that consistently generates structured CoT tokens that we can easily parse.

---

A conversation between User and Assistant. The user asks a question, and the Assistant solves it. The assistant first thinks about the reasoning process and then provides the user with the answer. User: Using the numbers [20, 14, 40], create an equation that equals 28. You can use basic arithmetic operations (+, -, *, /) and each number can only be used once. Show your work in <think> </think> tags. And return the final answer in <answer> </answer> tags, for example <answer> (1 + 2) / 3 </answer>.
Assistant: Let me solve this step by step. <think> We have the numbers 20, 14, and 40. We need to use these numbers to make an equation that equals 28 using basic arithmetic operations. Let's try different combinations:
40 - 14 - 20 = 10 - 20 = -10 (not 28)
40 - 14 + 20 = 26 + 20 = 46 (not 28)
40 + 14 - 20 = 54 - 20 = 34 (not 28)
40 + 14 + 20 = 54 + 20 = 74 (not 28)
40 * 14 / 20 = 560 / 20 = 28 **(this works)**
So, the equation that equals 28 is 40 * 14 / 20. </think> <answer> (40 * 14) / 20 </answer>

---

Our task, CountDown, is a simple testbed frequently used to study recent reasoning models Gandhi et al. (2024; 2025); Yao et al. (2023); Qin et al. (2025) – given a set of 3 or 4 operands (e.g., 19, 36, 55, 7) and target number (e.g., 65), the task is to find the right arithmetic combination of the operands to reach the target number (i.e., 55 + 36 - 7 - 19).

The model is given two rewards: accuracy reward for reaching the correct final answer, and a format reward when it generates its CoT tokens in between "<think>, </think>" tokens. For more details on how R1-Zero is trained, see (Guo et al., 2025). We refer to our task-specific model as $\text{R1}_{\text{Down}}^{\text{Count}}$.

One advantage of studying a specific task is in that preference training leads to mode collapse (Kirk et al., 2024; Padmakumar & He, 2024; Murthy et al., 2024; Slocum et al., 2025), resulting in a reduction in generation diversity. In our context, this is desirable, as the model converges to generating a highly structured CoT sequence. See Table 1.

This allows us to easily parse the model's CoT. Namely, the model enumerates through many attempts, while always marking each attempt as either "(this works)" or "(not {ans})". Thus, we can study the model's hidden states at specific timesteps, such as right before it produces either "this" or "not", which we refer to as $t_{valid}$ and $t_{invalid}$. We refer to the hidden states at these timesteps as $\mathbf{x}_{\text{Valid}}$ and $\mathbf{x}_{\text{Invalid}}$. We refer to the timestep in the prompt at which the target number is specified as $t_{ans}$.

## 4 COMPONENTS FOR SELF-VERIFICATION IN COUNTDOWN

Here we present a series of analyses to identify weights and subspaces relevant for verification. We do a "top-down" analysis to find relevant GLU vectors in late layers, and a "bottom-up" analysis to find relevant attention heads in early layers. Our analyses meet in the middle, to identify relevant subspaces for verification. We verify the role of such weights and subspaces via causal experiments.

### 4.1 TOP-DOWN: FINDING VERIFICATION-RELATED GLU VECTORS

**LogitLens.** We start our analysis by applying LogitLens (Nostalgebraist, 2020) to compare the hidden states of $\mathbf{x}_{\text{Valid}}$ and $\mathbf{x}_{\text{Invalid}}$ on a sample size of 300. We apply the unembedding layer at all intermediate layers $\mathbf{x}^{\ell}$ and inspect the resulting nearest neighboring tokens across 300 samples.

Figure 1(a, b) shows our results in the late layers (see Appendix Figure 5 for more layers). Interestingly, we see tokens such as "SUCCESS", "yes", "bingo" show up for $\mathbf{x}_{\text{Valid}}$, and "不符合" ("Does not conform"), "not", "不合" ("Incompatible") for $\mathbf{x}_{\text{Invalid}}$. Interestingly, we often observe English tokens for $\mathbf{x}_{\text{Valid}}$ and Chinese tokens for $\mathbf{x}_{\text{Invalid}}$. What drives these tokens to appear?

**(a) Orig LogitLens ("this")**

| | 1 | 2 | 3 | 4 |
|---|---|---|---|---|
| L 36 | this (0.98) | yes (0.02) | This (0.00) | works (0.00) |
| L 35 | this (0.86) | yes (0.12) | this (0.01) | works (0.00) |
| L 34 | this (0.74) | yes (0.21) | this (0.01) | yes (0.01) |
| L 33 | this (0.50) | yes (0.40) | yes (0.03) | works (0.01) |
| L 32 | yes (0.47) | yes (0.27) | this (0.10) | Yes (0.05) |
| L 31 | yes (0.64) | yes (0.34) | Yes (0.01) | _yes (0.00) |

**(b) Orig LogitLens ("not")**

| | 1 | 2 | 3 | 4 |
|---|---|---|---|---|
| L 36 | not (0.25) | (0.18) | 2 (0.05) | 3 (0.05) |
| L 35 | not (0.22) | (0.18) | 2 (0.04) | 3 (0.04) |
| L 34 | not (0.24) | (0.15) | = (0.04) | ( (0.04) |
| L 33 | not (0.21) | (0.16) | = (0.04) | ( (0.04) |
| L 32 | (0.15) | not (0.13) | Not (0.04) | = (0.03) |
| L 31 | not (0.14) | (0.11) | 这不是 (0.07) (This is not) | = (0.03) |

**(c) Intervene GLU Neurons: "this" to "not"**

| | 1 | 2 | 3 | 4 |
|---|---|---|---|---|
| L 36 | this (0.54) | yes (0.31) | works (0.04) | not (0.03) |
| L 35 | yes (0.42) | this (0.37) | not (0.03) | yes (0.02) |
| L 34 | yes (0.48) | this (0.20) | works (0.07) | yes (0.06) |
| L 33 | yes (0.35) | yes (0.30) | SUCCESS (0.07) | 符合条件 (0.06) (Qualifies) |
| L 32 | yes (0.67) | yes (0.29) | Yes (0.02) | 符合条件 (0.00) (Qualifies) |
| L 31 | SUCCESS (0.24) | success (0.17) | success (0.11) | bingo (0.06) |

**(d) Intervene Attn Heads: "this" to "not"**

| | 1 | 2 | 3 | 4 |
|---|---|---|---|---|
| L 36 | not (0.21) | (0.18) | 2 (0.04) | 3 (0.04) |
| L 35 | (0.15) | not (0.15) | = (0.04) | ( (0.04) |
| L 34 | (0.16) | not (0.11) | ( (0.04) | = (0.04) |
| L 33 | (0.15) | 不符 (0.05) (Does not match) | too (0.05) | = (0.03) |
| L 32 | (0.11) | 不符 (0.08) (Does not match) | 不满意 (0.05) (Not satisfied) | = (0.03) |
| L 31 | not (0.04) | 不符合 (0.04) (Does not conform) | 不合 (0.03) (Incompatible) | 不满意 (0.03) (Not satisfied) |

Top K

Figure 1: **Averaged LogitLens from 300 samples.** We see tokens related to verification ("success", "不合") in the last few layers. **(a), (b)** show the top tokens when (in)correct solutions are reached. **(c), (d)** shows results from intervening on either GLU weights or attention heads, given a correct solution. For **(c)**, while the model is less certain (P("this") drops from 0.98 to 0.54), we still see tokens such as "success" showing up. For **(d)**, we no longer see any tokens related to "success", and the model's final next-token predictions closely resembles when the model has not found a solution (b).

**Probing.** To answer this question, we train linear probes $W^\ell \in \mathbb{R}^{2 \times d}$ at every layer $\ell$ from timesteps right before "this" or "not" is predicted. These timesteps correspond to when an answer is produced, and an open parenthesis tokens "(" is being predicted next, as opposed to "this" or "not".

$W^\ell$ is a linear mapping from the hidden states, $\mathbf{x}^\ell$, to a binary label of whether the model has found the solution. Our training data is $\mathcal{D} = \{(\mathbf{x}^\ell_{y^i}, y^i)\}_{i=0}^{N-1}, y^i \in \{0 \text{ ("not")}, 1 \text{ ("this")}\}$, N=327,680. We solve for $W^\ell$ to fit $y = \text{softmax}(W^\ell \mathbf{x}^\ell)$ using gradient descent (hyperparameters in Appendix C).

Validation accuracy ($N = 512$) per layer is provided in the Appendix (Figure 6), with accuracy usually staying above 90% after the first few layers. High accuracy suggests that our probing vectors $W[0], W[1]$ identify a direction in the model's activation space that linearly separates points of $\mathbf{x}_{\text{Valid}}$ and points of $\mathbf{x}_{\text{Invalid}}$ (i.e., linearly separable subspaces).

Such vectors can steer the model. Simply adding $W[0]$ or $W[1]$ into hidden states can push $\mathbf{x}$ towards $\mathbf{x}_{\text{Valid}}$ or $\mathbf{x}_{\text{Invalid}}$, and change the model's output to indicate that it has (or has not) found a solution, even when it has not (or has). We provide qualitative examples of steering results in Appendix E.

**GLU$_{\text{Valid}}$, GLU$_{\text{Invalid}}$ Vectors.** Our probe $W$ tells us that mid-layer activations can be linearly separated to identify solved cases ($\mathbf{x}_{\text{Valid}}$) from unsolved cases ($\mathbf{x}_{\text{Invalid}}$), but also serves a secondary purpose. Namely, we can use $W$ to identify GLU$_{\text{Out}}$ vectors of interest (Lee et al., 2024).

Per layer, we select the top $k(= 50)$ GLU$_{\text{Out}}$ vectors by how similar they are to $W^\ell[0]$ or $W^\ell[1]$ using cosine similarity. One can consider these vectors as weights that contribute the most towards $W^\ell[0]$ (no solution) or $W^\ell[1]$ (found solution) directions. We refer to them as GLU$_{\text{Invalid}}$ and GLU$_{\text{Valid}}$ vectors. This results in $k \times L \times 2$ GLU$_{\text{Valid, Invalid}}$ vectors (0.9% of the model's GLU$_{\text{Out}}$ vectors).

Unembedding GLU$_{\text{Valid/Invalid}}$ vectors reveal which tokens get promoted when they are activated. Table 2 shows their nearest neighbors in the model's token embedding space. We observe that most interpretable GLU$_{\text{Valid/Invalid}}$ neurons occur in the second half of layers. Interestingly, we again note that there seems to be a correlation between GLU$_{\text{Valid/Invalid}}$ and English versus Chinese tokens, hinting at the underlying geometry of $\mathbf{x}_{\text{Valid/Invalid}}$ and the model's embedding space.

Table 2: **GLU$_{\text{Out}}$ vectors relevant to verification, and their nearest neighbors.** $W[0], W[1]$ indicate our probe model. "$(x, y)$" indicates the GLU$_{\text{Out}}$ vector at layer $x$, index $y$. "$-1 \times (x, y)$" (marked in red) indicates the antipodes of the GLU$_{\text{Out}}$ vector at layer $x$, index $y$. Interestingly, we observe a correlation between valid/invalid vectors and English and Chinese.

| Vector | Nearest Neighbors |
|---|---|
| $W[0]$ | 不完 (unfinished), 不了 (unable), 不 (not), 不在 (absent), 不该 (should not) |
| $W[1]$ | Exactly, >(, =yes, =YES, =:, ===, quis, esac, #### |
| (26, 744) | 未能 (failed), 不够 (not enough), nicht (not), 不像 (not like), 达不到 (can't reach) |
| (26, 6619) | 缺乏 (lack), 缺少 (lack), 不方便 (inconvenient), lacks, 难以 (difficult), 未能 (failed) |
| (27, 4971) | inefficient, 没能 (failed), 不方便 (inconvenient), Danger, disadvantage, 不利于 |
| (29, 6676) | yes, Yes, Bindable, exactly, Yes, "Yes, yes, Yep, Exactly, included |
| (27, 10388) | mirac, 乐观 (optimism), 安然 (safely), Relief, 幸 (fortunate), .isSuccess |
| (30, 8233) | correctly, 正确 (correct), 恰当 (appropriate), accurately, 符合 (conform) |
| -1×(26, 744) | 慎 (careful), 足 (sufficient), 同等 (equal), tend, ONDON, 足以 (enough) |
| -1×(26, 6619) | 不仅能 (not only can), 不错的 (good), 具有良好 (have good), 总算 (finally) |
| -1×(27, 4971) | successfully, successful, 顺利 (smooth), 成功 (successful), 删除成功 |
| -1×(29, 6676) | 都不 (neither), 不太 (not quite), neither, 不予 (not given), 没见过 (never seen) |
| -1×(27, 10388) | 失败 (failure), failure, 不良 (bad), 不利 (unfavorable), 糟糕 (bad), 失误 (mistake) |
| -1×(30, 8233) | wrong, 不良 (bad), incorrect, wrong, invalid, bad, inappropriate, invalid |

While GLU$_{\text{Valid/Invalid}}$ encode verification-related tokens, what role do they play? This can be partially answered by applying LogitLens again on 300 samples, but now by "turning off" GLU$_{\text{Valid}}$ vectors (< 1% of total GLU vectors) by scaling them to zero. Figure 1(c) shows the results: while the probability of verification-related tokens drop (e.g., P("this") drops from 0.98 to 0.70 in layer 36), the end behavior remains the same (i.e., "this" is still the top-1 token). This tells us that GLUs do not fully explain self-verification. We demonstrate a more thorough causal analysis in Section 4.4.

## 4.2 BOTTOM-UP: PREVIOUS-TOKEN ATTENTION HEADS FOR VERIFICATION ($\mathbf{A}_{\text{PREV}}$)

We next inspect the role of attention heads for verification. One motivation for choosing CountDown as our task is that the task specifies the target number in the context. Thus we can posit that a Transformer could verify its CoT tokens by comparing them against the specified target number (at timestep $t_{ans}$). Such a hypothesis provides an entry way for our bottom-up analysis.

We test our hypothesis by inspecting the attention patterns whenever the model's CoT produces the correct answer. We filter for attention heads that spend at least 10% of its attention on $t_{ans}$, and refer to these as previous-token heads (notated $\mathbf{A}_{\text{Prev}}$). Previous-token heads are not new: they were first discussed in the context of induction heads Olsson et al. (2022). We identify 33 previous-token heads (out of a total of 576 heads). Interestingly, we find that most previous-token heads occur roughly in the first half layers (except for one at layer 31, all are at or before layer 22). In Section 4.4 we demonstrate via causal interventions that disabling previous-token heads can disable model verification. But first, what is the relationship between GLU$_{\text{Valid/Invalid}}$ vectors and $\mathbf{A}_{\text{Prev}}$ heads? Below we adapt inter-layer component channels to understand their relationship.

## 4.3 JOINING GLU$_{\text{VALID}}$ AND $\mathbf{A}_{\text{PREV}}$: IDENTIFYING VERIFICATION SUBSPACES (POLYTOPES)

We identify subspaces for self-verification by studying the relationship between GLU$_{\text{Valid}}$ vectors and $\mathbf{A}_{\text{Prev}}$ attention heads. As a reminder, we observe that $\mathbf{A}_{\text{Prev}}$ usually occurs in the first half layers (1 to 22), while GLU$_{\text{Valid}}$ vectors usually occur in the later half (18 to 36). We hypothesize and empirically verify that $\mathbf{A}_{\text{Prev}}$ activates GLU$_{\text{Valid}}$ vectors.

First, we borrow from neuroscience to define *receptive fields* (Olshausen & Field, 1997). For a single neuron $k$ that computes an activation function $f^k : \mathbb{R}^d \to \mathbb{R}$, its receptive field is defined as

$$S_k = \left\{ \mathbf{x} \in \mathbb{R}^d \mid f^k(x) > 0 \right\} \tag{6}$$

In simpler terms, $S_k$ is the subspace that triggers a neuron active. In the context of GLUs, this means

$$S_k = \left\{ \mathbf{x} \in \mathbb{R}^d \mid \phi(W_{gate}^k \mathbf{x}) \cdot W_{up}^k \mathbf{x} > 0 \right\} \tag{7}$$

Now consider a set of neurons, $K$, and the intersection of all of their receptive fields: $\mathbf{S}_K = \bigcap_i^{|K|} S_i$.

$\mathbf{S}_K$ can be considered a polytope in the model's activation space parameterized by $\left\{ W_{gate}^i, W_{up}^i \right\}_{i=1}^{|K|}$. Here we use the receptive fields of $\text{GLU}_{\text{Valid}}$ to further localize self-verification.

Namely, we identify a small subset of as few as six previous-token heads that disables self-verification. To do so, we look for previous-token heads (as opposed to hidden states) that activate $\text{GLU}_{\text{Valid}}$ neurons the most. The output of each head is its OV-circuit (i.e., $W_O W_V$), scaled by to some attention distribution. Meanwhile, the strength of each $\text{GLU}_{\text{Valid}}$ activation is determined by $W_{gate}$ and $W_{up}$.

We score each previous-token head $h$ by first decomposing its OV-circuit using SVD:

$$(W_O W_V)^h = U\Sigma V^\top, \Sigma = \text{diag}(\sigma_{h,1} \geq ... \geq \sigma_{h,k} \geq 0)$$

where $k = d^{head}$ because $W_O W_V$ is low-rank. Let $\mathbf{u}_{h,j}$ refer to the $j$-th left-singular vector.

Each head is then assigned the following score:

$$\text{score}(h, \text{GLU}_{\text{Valid}}) = \frac{1}{N} \sum_i^N \sum_j^k \sigma_{h,j}^2 \left( \phi(W_{gate}^i \mathbf{u}_{h,j}) \cdot W_{up}^i \mathbf{u}_{h,j} \right) \tag{8}$$

where $\text{GLU}_{\text{Valid}} = \{W_{gate}^i, W_{up}^i\}_{i=0}^{N-1}$, $N = |\text{GLU}_{\text{Valid}}|$, $W_{gate}^i, W_{up}^i, \mathbf{u}_{h,j} \in \mathbb{R}^d$.

Put differently, we decompose $W_O W_V$ into singular vectors, and measure how well they each align with the receptive fields of $\text{GLU}_{\text{Valid}}$ ($W_{gate}, W_{up}$), weighted by their singular values.

Note that we only need the left singular vectors because we only need the output subspace of $W_O W_V$, which is spanned by the left singular vectors. Also note that weighing by $\sigma_{h,j}^2$ can be thought of as distributing the Frobenius norm of $W_O W_V$ across each mode, as $||W_O W_V||_F^2 = \sum_i \sigma_i^2$.

Our score is akin to inter-layer communication channels (Elhage et al., 2021; Merullo et al., 2024), which scores how strongly two attention heads "talk" with each other using a "composition score":

$$CS(W_1, W_2) = \frac{\|W_1 W_2\|_F}{\|W_1\|_F * \|W_2\|_F} \tag{9}$$

where $W_1$ is an OV component of one head and $W_2$ the QK component of another at a later layer. Our formulation can be considered a composition score between attention heads and GLUs, using both $W_{gate}$ and $W_{up}$ in place of $W_2$ with some additional steps in between.

Once we score each previous-token head using Eq. 8, we incrementally ablate one head at a time until we achieve perfect intervention scores (Section 4.4). Using this approach, we identify as few as **six** attention heads that can disable model verification. We notate this subset as $\mathbf{A}_{\text{Verif}}$.

To summarize, we claim that the model has subspace(s) (polytope(s)), $\mathbf{S}_{\text{GLU}_{\text{Valid}}}$, for self-verification. The model's hidden state enters this subspace when it has verified its solution. In our setting, given the nature of our task, previous-token heads $\mathbf{A}_{\text{Prev}}$ take the hidden-state into this subspace, while for other tasks, different components may be used. This subspace also activates verification-related GLU weights, promoting the likelihood of tokens such as "success" to be predicted (Figure 3).

We find that alternative hyperparameters or scoring functions can yield different subsets of previous-token heads that also disable self-verification. We discuss these results in Appendix F. This suggests that we do not identify a full circuit, but rather a critical component for verification. Also note that our scoring function makes simplifications by ignoring possible interactive effects across heads, as well as transformations (layer norms, GLUs) across layers. Regardless, our finding remains robust: a small subset of previous-token heads can disable verification.

## 4.4 Causal Interventions

We study the role of each component above with causal intervention. Our test set consists of 300 samples in which the model originally correctly finds and validates its solutions. For each test case,

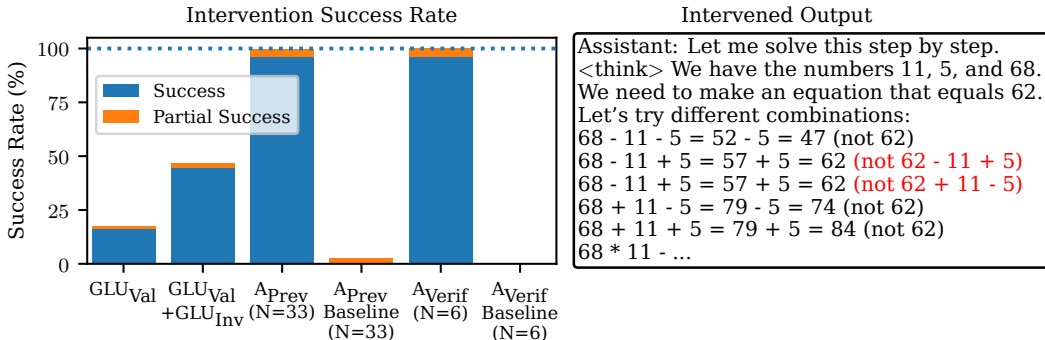

Figure 2: **Intervention Results: Disabling as few as 6 attention heads disables self-verification**, rendering the model to generate tokens indefinitely. $A_{Prev}$ refers to 33 previous-token heads. $A_{Verif}$ refers to a subset of 6 previous-token heads. $A_{Prev}$ **Baseline**, $A_{Verif}$ **Baseline** refer to the average of 5 runs, where each run randomly samples 33 (or 6) attention heads.

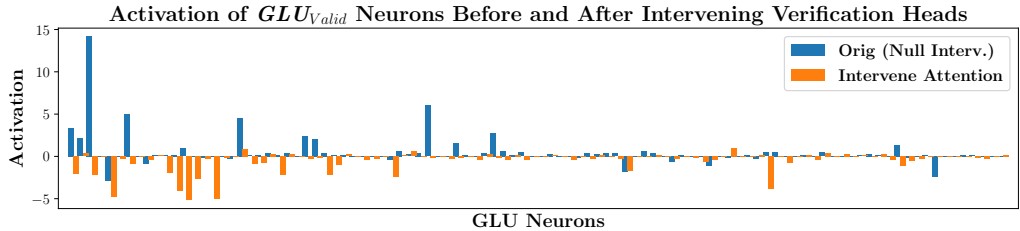

Figure 3: **GLU$_{Valid}$ activations before and after turning off 6 $A_{Verif}$ attention heads.** Adjacent pairs of blue and orange bars indicate the same GLU$_{Valid}$ vector. Turning off our identified attention heads leads to a significant drop in their activations.

the model generates 100 tokens. Every time an attempt for a solution is made (i.e., "(this works)" or "(not ({ans})" is about to be predicted), we turn off some of the model weights as described below. We measure intervention success rate: the percentage of times the model fails to validate its solution, despite having found the solution within the 100 generated tokens. Interestingly, the model occasionally marks a correct attempt as invalid (desired intervention result), but continues its generation to say it has found a solution (e.g., given target number 62, the model will output "68 - 11 + 5 = 62 (not 62) So, the answer is 68 - 11 + 5"). We mark these cases as partial successes.

We ablate a few components: (1) **GLU$_{Valid}$:** We zero-out 50 GLU$_{Valid}$ vectors per layer from layer 18 to 36 (the second half of the model; 0.45% of all GLU vectors). (2) **GLU$_{Valid}$ & GLU$_{Invalid}$:** We zero-out the top 50 GLU$_{Valid}$ and top 50 GLU$_{Invalid}$ vectors per layer (0.9% of all GLU vectors). (3) **A$_{Prev}$:** We turn off 33 previous-token heads (5.7% of all attention heads) by zeroing-out their $W_O$ weights. (4) **A$_{Verif}$:** We turn off **6** verification heads identified via Eq. 8: L17H14, L17H11, L17H10. We also provide two baselines: **A$_{Prev}$ Baseline** and **A$_{Verif}$ Baseline**, which each report the average from 5 runs, where each run randomly samples 33 (or 6) attention heads.

Results are shown in Figure 2, with an example of an intervened output. We achieve near perfect intervention rates when turning off 33 previous-token heads, or a smaller subset of 6 heads. The model misclassifies correct solutions as invalid without these heads, and continue its CoT indefinitely.

These attention heads directly affect the GLU$_{Valid}$ vectors. Figure 3 demonstrates the activations of the top GLU$_{Valid}$ vectors before and after disabling the subset of 6 previous-token heads. In most cases, we observe a large drop (to near 0, or often even negative values) in GLU$_{Valid}$ activations.

**Antipodal GLU$_{Out}$ Vectors.** While attention heads achieve near perfect interventions, Figure 2 also indicates that disabling both GLU$_{Valid}$ and GLU$_{Invalid}$ performs better than disabling just GLU$_{Valid}$. Why should disabling GLU$_{Invalid}$ improve intervening, i.e., make the model fail at verification?

This can be explained by two facts: (1) the geometry of GLU$_{Valid}$ and GLU$_{Invalid}$ vectors, and (2) the nonlinear activation used in GLU. Interestingly, we find that the antipodal directions of GLU$_{Valid}$ and GLU$_{Invalid}$ also often encode tokens relevant for verification. The last 7 rows of Table 2 marked in red

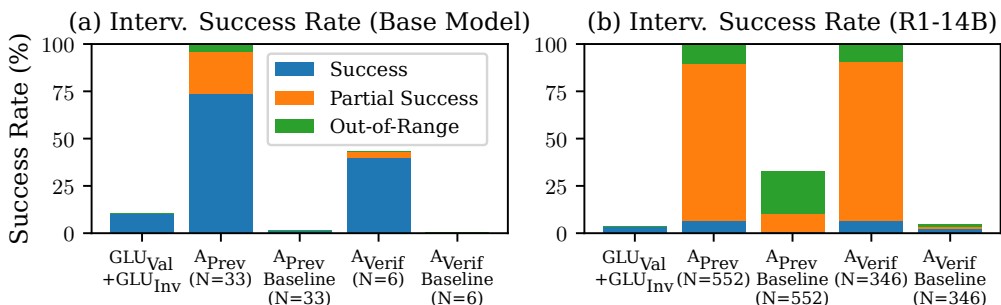

Figure 4: **Intervention Results for the base model and $R1_{14B}$.** In the base model, $A_{Prev}$ can similarly disable self-verification, while $A_{Verif}$ only plays a partial role for verification, hinting at the effects of RL on their weights. In $R1_{14B}$, interventions mostly leads to partial success, in which the model first marks a solution as incorrect but self-corrects itself, hinting at a larger verification circuit. Also interestingly, the smaller subset of $A_{Verif}$ is more effective at self-verification than $A_{Prev}$.

indicate the nearest neighbors of the antipodes of $GLU_{Valid}$ and $GLU_{Invalid}$. In addition, Qwen2.5-3B uses SiLU activations (Hendrycks & Gimpel, 2016). Thus inactive neurons take on small *negative* values (as opposed to zero, had ReLU been used).

With that said, consider only zeroing out $GLU_{Valid}$ neurons: given a correct CoT sequence, $GLU_{Invalid}$ vectors are inactive. However, because of SiLU, the inactive $GLU_{Invalid}$ vectors have negative activations, thus get multiplied by a small *negative* value, flipping directions, and therefore contribute towards the "success direction". In the case of zeroing out both $GLU_{Valid}$ and $GLU_{Invalid}$, we are further zeroing out the effects of inactive $GLU_{Invalid}$ neurons.

## 5 SIMILAR VERIFICATION COMPONENTS IN BASE MODEL & DEEPSEEK-R1

We verify that similar verification subspaces exist in our base model (Qwen2.5-3B), as well as a general reasoning model, DeepSeek-R1-Distill-Qwen-14B (henceforth $R1_{14B}$).

For both models, we provide CountDown as an in-context learning (ICL) task, including 5 demonstrations of solving CountDown using the structured output of $R1_{Down}^{Count}$. We find that both models can solve the ICL version of CountDown while following the same CoT structure of $R1_{Down}^{Count}$, corroborating recent findings that reasoning capabilities often already exist in pre-trained models (Yue et al., 2025; Gandhi et al., 2025).

We then repeat our intervention analyses above. In our ICL setting, our interventions sometimes make the model generate "out-of-range", by which we mean their generations do not adhere to the structured CoT of $R1_{Down}^{Count}$. We mark these cases as out-of-range.

**Base Model.** Figure 4 (a) shows the interventions from Section 4.4 on our base model. Previous-token heads still achieve near perfect (partial) intervention rates, suggesting a similar mechanism for self-verification in the base model. Also note that the six $\mathbf{A}_{Verif}$ heads demonstrate a lower success rate. Similar to Prakash et al. (2024), which demonstrates that fine-tuning enhances existing mechanisms in a base model, we hypothesize that RL enhances an existing verification mechanism, thus resulting in highly localized attention heads in $R1_{Down}^{Count}$ that can control self-verification.

**DeepSeek-R1 Model.** In the case of $R1_{14B}$, we repeat the procedures in Sections 4.1~4.4. However, a probe vector $W$ is required to identify $GLU_{Valid/Invalid}$. Thus we apply EMB2EMB (Lee et al., 2025), a simple technique to transfer and re-use steering vectors across language models (see Appendix G for a brief explanation). Applying EMB2EMB on $R1_{Down}^{Count}$'s probe, $W$, results in a probe vector $W_{R1}$ for $R1_{14B}$, allowing us to repeat our analyses from Section 4.1 on $R1_{14B}$.

We find similar $GLU_{Valid, Invalid}$ vectors in $R1_{14B}$, analogous to Table 2 (see Appendix H), hinting at similar verification subspaces in $R1_{14B}$. We identify and intervene on previous-token heads ($A_{Prev}$) in $R1_{14B}$, following Section 4.2. We use an attention threshold of 5% (as opposed to 10% in $R1_{Down}^{Count}$)

to compensate for the longer context induced from our ICL setup, which yields 552 (out of 1920) previous-token heads. We discuss results from different hyperparameters (thresholds) in Appendix I.

We also replicate Section 4.3 to identify a smaller subset of 351 attention heads that achieve near perfect (partial) intervention success rates.

Results are shown in Figure 4 (b). Interestingly, our interventions mostly lead to partial successes in $R1_{14B}$, in which the model initially fails at self-verification (labels a correct solution as "(not {ans})"), but corrects itself (generates "Wait, 68 - 11 + 5 is 62 so that works."). This hints at a larger verification circuit for $R1_{14B}$. We also note that $\mathbf{A}_{Verif}$ has a higher success rate than $\mathbf{A}_{Prev}$, despite being a smaller set, suggesting that not all previous-token heads (or their interactions) are helpful in self-verification. We leave further exploration for future work.

# 6 RELATED WORK

**Decoding Interpretable Representations.** A growing line of work focuses on decoding and manipulating interpretable representations in model activations (Zou et al., 2023). Conveniently, many concepts take on *linear* representations (Mikolov et al., 2013; Nanda et al., 2023; Park et al.), in which simple vectors encode human-interpretable concepts.

For "non-reasoning" models, researchers have studied "truthful" representations before (Burns et al.), where steering towards a "truthful" direction has led to improvements in tasks related to factual recall (Li et al., 2023). In a similar vein, researchers have shown that the model's representations can reveal whether they will make errors (e.g., hallucinations) (Orgad et al., 2024), or when they are unable to recall facts about an entity (Ferrando et al., 2024).

Most recently, concurrent work also investigate how models solve reasoning tasks. Zhang et al. (2025) find that models know when they have reached a solution, while Venhoff et al. (2025) decode directions that mediate behaviors such as handling uncertainty or self-corrections. In our work, we take a deeper dive into how a reasoning model verifies its own reasoning trace.

**Circuit Analysis.** A growing line or work decomposes the forward pass of a neural network as "circuits" (Olah et al., 2020), or computational graphs. This allows researchers to identify key components and their causal effects for a given forward pass. A common approach to construct computational graphs is to replace model components with dense activations with a sparsely-activating approximation. Dunefsky et al. introduces Transcoders to approximate MLP layers, while Ameisen et al. (2025) further develops Cross-layer Transcoders to handle inter-layer features. Lindsey et al. (2025) uses Cross-layer Transcoders to conduct circuit analyses for a wide range of behaviors, such as multi-step reasoning (for factual recall) or addition, and also investigate when a model's CoT is (un)faithful. In our work, we identify key components needed for a potentially larger verification circuit without the need for separate sparse approximations.

# 7 DISCUSSION

We studied how a task-specific model verifies its own outputs. We repurposed mode collapse as a feature, not a bug: by leveraging the fact that preference tuning leads to mode collapse, we train a model with highly structured CoT, making it easy to parse its reasoning trace. With this setup, we found GLU weights that encode verification-related tokens, and previous-token heads that can disable verification. We offer a simple extension to inter-layer communication channels that allow us to localize as few as six attention heads that can also disable verification. Finally, we verify the existence of similar components in our base model and a general reasoning DeepSeek-R1 model. We view our work as a step towards understanding the inner mechanisms of recent reasoning models.

**Limitations.** Note that we do not claim to have uncovered a full verification circuit, but rather critical components for verification. We also reiterate the scope of our work: we study a specific task that allows for **context-based verification**. Obviously, not all reasoning tasks share this property: many tasks likely require **prior-based verification** using general knowledge. We speculate that similar subspaces are used for prior-based verification, but is less obvious where they show up.

## 8 REPRODUCIBILITY STATEMENT

Our code to reproduce all of our experiments can be found in the attached supplementary files. Appendix A, C, I provide details of our training setup and hyperparameters.

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

## A    HYPERPARAMETERS FOR R1

Here we provide the hyperparameters used to train $\text{R1}^{\text{Count}}_{\text{Down}}$.

| Parameter | Value |
|---|---|
| Train Batch Size | 256 |
| Validation Batch Size | 1312 |
| Max Prompt Length | 256 |
| Max Response Length | 1024 |
| Actor Learning Rate | 1e-6 |
| PPO Mini Batch Size | 128 |
| PPO Micro Batch Size | 8 |
| Log Prob Micro Batch Size | 8 |
| Tensor Model Parallel Size | 2 |
| Critic Learning Rate | 1e-5 |
| KL Coefficient | 0.001 |

Table 3: Training Hyperparameters.

## B    LOGITLENS ON MORE LAYERS

Figure 5 demonstrates LogitLens as described in Section 4.1 on more layers.

## C    HYPERPARAMETERS FOR PROBING

We use a batch size of 8, validation size of 256, weight decay of 0.01, and learning rate of 1e-4. We validate every 50 gradient steps, and terminate training when validation loss has not improved after a patience value of 10.

## D    PROBE ACCURACY

Figure 6 shows probing results. The model has a linear separation in its hidden states given correct versus incorect CoT tokens.

## E    EXAMPLES OF STEERING VERIFICATION WITH PROBE

Once we identify a direction that encodes solved versus unsolved states (i.e., $W_{probe}$), we can simply add this direction into the model's hidden states to make the model believe that it has found a solution:

$$\mathbf{x}^\ell = \mathbf{x}^\ell + \alpha W_{probe} \tag{10}$$

where $\mathbf{x}^\ell, W_{probe} \in \mathbb{R}^d$ and $\alpha \in \mathbb{R}$. Some hyperparameters include $\ell$ (which layers to steer on) and $\alpha$, where a larger $\alpha$ amplifies the target behavioral effect.

While an extensive hyperparameter search and a systematic experiment may be useful, steering is not a core component but rather a tangential experiment. We thus provide qualitative examples using $\ell = \{n \mid 24 \le n \le 36\}$ and $\alpha = 20$ (after normalizing $W_{probe}$) in Table 5.

## F    ALTERNATIVE SUBSETS OF PREVIOUS-TOKEN HEADS

Of the 25 previous-token heads that we identify, there are many ways to identify subsets that disable verification. We offer a few examples, and document how many heads are needed to disable verification with perfect success rates (including partial successes).

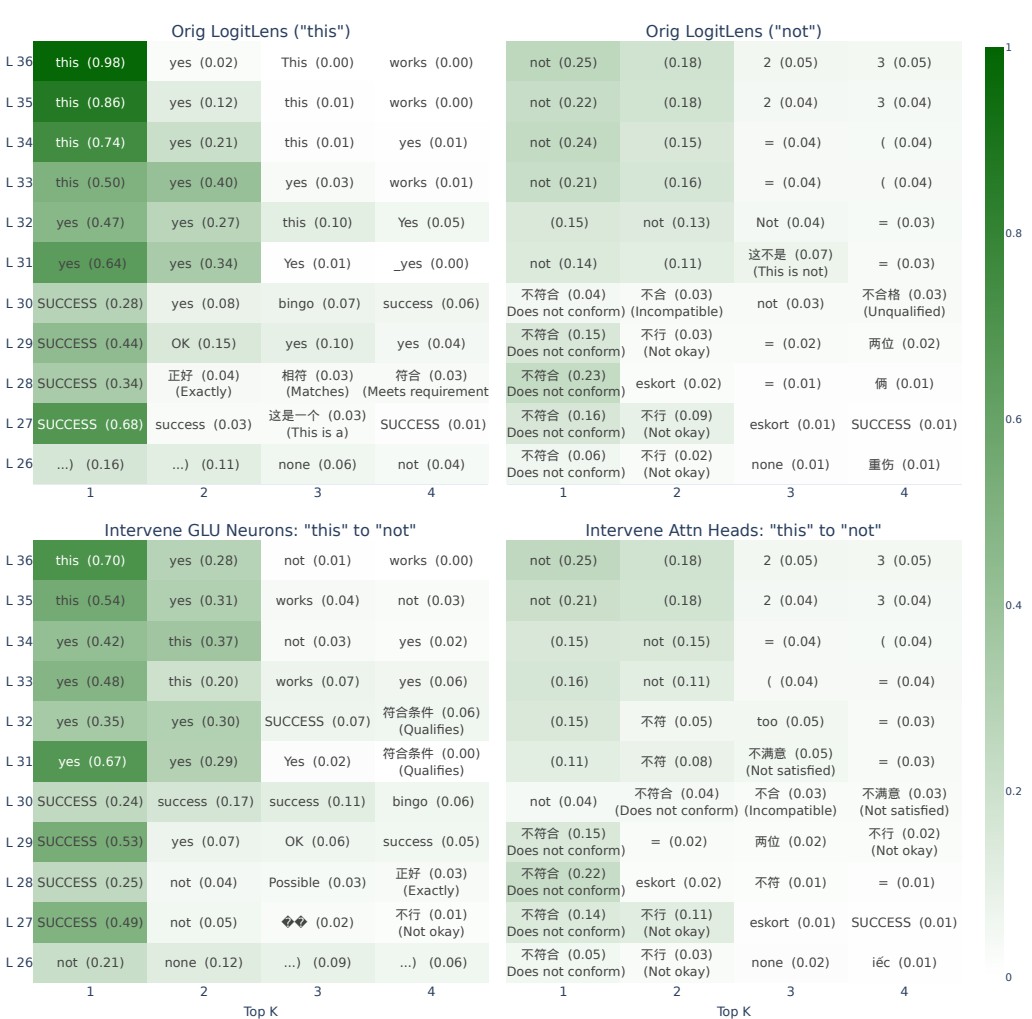

Figure 5: **Averaged LogitLens from 300 samples** (Same as Figure 1 but demonstrating more layers). We see tokens related to verification ("success", "incorrect") in the last few layers. **(A), (B)** show the top tokens when a correct / incorrect solution is reached. **(C), (D)** shows results from intervening on either GLU weights or attention heads, given a correct solution. For **(C)**, while the model is less certain (P("this") versus P("not") becomes 0.51 vs. 0.49 in last layer), we still see tokens such as "success" showing up. For **(D)**, we no longer see any tokens related to "success" show up, and the model is certain that it has not found a solution.

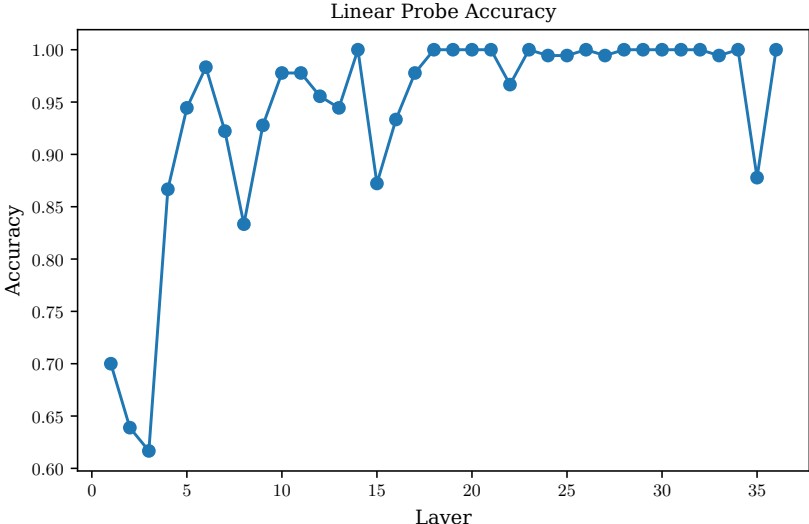

Figure 6: **Probe Accuracy.** The model has a linear separation in its hidden states given correct versus incorrect CoT tokens.

**Attention Density.** The simplest method is to sort the heads based on how much they attend to the target token that timestep $t_{ans}$.

**Varying Hyperparameters.** Given our approach (Section 4.3), we try different parameters for N.

**Sort by Similarity to $W_{gate}, W_{up}$.** An alternative approach is to sort the attention heads based on how similar their OV circuits are to the $W_{gate}$ and $W_{up}$ weights of the $\text{GLU}_{Valid}$ weights. We simply take the mean of $\{W^i_{gate}, W^i_{up}\}^N_i$ vectors from our N $\text{GLU}_{Valid}$ weights. We then sort attention heads based on how similar they are to the resulting mean vector.

**Sort by Probe $W_{Probe}$.** An alternative is to sort the attention heads based on how similar their OV circuits are to the probe vector $W_{Probe}$.

Table 6 demonstrates how many heads are needed from each approach to disable verification with perfect accuracy.

## G BRIEF EXPLANATION OF EMB2EMB

Language models represent numerous concepts using *linear* representations (Nanda et al., 2023; Park et al.), by which we mean one can add a single vector that encodes a specific concept into the activations during inference-time to raise the probability for the model to exhibit such concept or behavior (Rimsky et al., 2024; Lee et al., 2024; Li et al., 2023). Researchers often refer to such vectors as "steering vectors".

In other words, during the forward pass, imagine the activations at layer $i$. One can simply add a steering vector $W$ (scaled by some hyperparameter $\alpha$) to control the model's behavior:

$$\mathbf{x}^{i+1} = \mathbf{x}^i + F^i(\mathbf{x}^i) + \alpha W \tag{11}$$

where $\mathbf{x}^i$ and $F^i$ are the hidden state and transformer block at layer $i$.

EMB2EMB (Lee et al., 2025) is a simple method that transfers a steering vector from one language model to another, by leveraging the fact that the unembedding spaces of language models are often quite similar.

EMB2EMB works as following. Given a "source" and "target" language model, $\mathcal{M}_S$ and $\mathcal{M}_T$, first randomly sample a set of $N$ (= 100,000) tokens, noting their token (un)embeddings as $\mathcal{E}_S$ and $\mathcal{E}_T$.

| Vector | Nearest Neighbors |
|---|---|
| (36, 10079) | 不失 (not losing), NotNull, 得起 (can afford), 得住 (can endure), 不惜 (not hesitate) |
| (32, 497) | 删除成功 (deletion successful), successes, Success, success, succeeded, favorable |
| (35, 6041) | 的强大 (powerful), excellent, powerful, 强大的 (powerful), 很棒 (great), strong, 优异 |
| (37, 5399) | 等于(equal), equal, 同样的(same), 相同(same), equals, 相同的 (same), 同等 (equal) |
| (32, 13572) | successfully, 成功 (success), 解决了 (solved), 实现了 (achieved), 顺利 (smoothly) |
| (30, 10150) | 没问题 (no problem), 无忧 (no worries), .NoError, harmless, 不变 (unchanged) |
| (45, 6650) | 没有 (do not have), 不存在 (does not exist), 没有任何 (do not have any), 不需要 |
| (39, 6070) | never, 不会 (will not), doesn, not, 不能 (cannot), nowhere, cannot, neither |
| (46, 12380) | neither, none, nowhere, None, Neither, none, nobody, cannot |
| (44, 12793) | não (not), 不 (not), nicht (not), tidak (no), н е (not), ikke (not), niet (not) |
| (41, 12498) | 不在 (not present), 不再 (no longer), non, 非 (non-), 不再是 (is no longer), 不属于 |
| (37, 7636) | 不合适 (inappropriate), 不足 (insufficient), 达不到 (cannot reach), 不够 (not enough) |
| (31, 5164) | 没能 (did not), fails, 未能 (failed), 不够 (not enough), 做不到 (cannot), 不及 |
| (35, 2509) | 不 (not), 不含 (does not contain), 不对 (incorrect), 不影响 (does not affect), |

Table 4: **GLU$_{\text{Out}}$ vectors relevant to verification in R1$_{\text{14B}}$.**

Then, learn a linear transformation, $T$, to map points $\mathcal{E}_S$ to $\mathcal{E}_T$, using something as simple as least squares minimization. Note that $T$ maps between spaces with different dimensions.

Given transformation $T$ and a steering vector $W_S$ from the source model $\mathcal{M}_S$, one can steer the target model $\mathcal{M}_T$ by simply applying transformation $T$ to $W_S$:

$$\mathbf{x}_T^{i+1} = \mathbf{x}_T^i + F_T^i(\mathbf{x}_T^i) + \alpha T W_S, \tag{12}$$

where $\mathbf{x}_T$ is the activations and $F_T$ is the transformer block of target model $\mathcal{M}_T$. In our work, we use EMB2EMB to transfer our probe vector $W$ from R1$_{\text{Down}}^{\text{Count}}$ to a general reasoning R1 model, R1$_{\text{14B}}$.

## H GLU$_{\text{VALID, INVALID}}$ IN DEEPSEEK-R1-DISTILL-QWEN-14B

Table 4 contains GLU$_{\text{Out}}$ weights in R1$_{\text{14B}}$ related to verification.

## I ALTERNATIVE HYPERPARAMETERS FOR R1

We add a quick note on using alternative hyperparameters for the experiment on R1$_{14B}$ in Section 5. Namely, there are two hyperparameters, the attention threshold to categorize an attention head as a previous-token head ($A_{\text{Prev}}$), and the number of GLU$_{\text{Valid}}$ vectors to include to identify a smaller subset of attention heads ($A_{\text{Verif}}$).

For the first hyperparameter (attention threshold), we use 5% as our threshold, as reported in the main text. We find that using a higher threshold (10%) is too constrained and perhaps yields too small of a set of previous-token heads, leading to a lower intervention success rate (0.4%, with 25% partial success rate). We believe this is because there are still many attention heads that attends to the correct solution (say, with an attention pattern < 10% but > 5%) to offset our intervention.

Using a smaller threshold (2.5%) yields too large of a set of previous-token heads. In this case, interventions now lead to a much higher rate of "out-of-range" results – i.e., the model no longer adheres to the structured CoT sequence after our interventions.

Regarding the number of GLU$_{\text{Valid}}$ vectors, we see our results converge to the same results after 200 vectors, which is what we use in our experiments.

## J LLM USAGE

We used LLMs to proof read our draft and polish our notations.

| Operands | Target | Type | Output |
|---|---|---|---|
| 3, 87, 38 | 46 | Orig. | \<think\>
87 - 38 - 3 = 116 - 3 = 113 (not 46)
87 - 38 + 3 = 49 + 3 = 52 (not 46)
87 - 38 + 3 = 49 + 3 = 52 (not 46)
87 + 38 - 3 = 125 - 3 = 122 (not 46)
38 + 38 - 3 = 76 - 3 = 73 (not 46)... |
| 3, 87, 38 | 46 | Steer | \<think\>
87 - 38 - 3 = 116 - 3 = 113 (not 46)
87 - 38 + 3 = 49 + 3 = 52 (not 46)
87 - 38 + 3 = 49 + 3 = 52! Yes, we got it!
87 - 38 + 3 = 49 + 3 = 52.
\</think\>
\<answer\> 87 - 38 + 3 \</answer\> |
| 60, 62, 12, 34 | 68 | Orig. | \<think\>
60 - 62 + 12 + 34 = 54 + 12 + 34 = 66 + 34 = 100 (not 68)
60 + 62 - 12 - 34 = 122 - 12 - 34 = 108 - 34 = 74 - 34 = 40 (not 68)
60 + 62 - 34 + 12 = 122 - 34 + 12 = 88 + 12 = 100 - 34 = 66 (not 68)
60 + 34 + 12 - 62 = 94 + 12 - 62 = 106 - 62 = 44 (not 68)... |
| 60, 62, 12, 34 | 68 | Steer | \<think\>
60 - 62 + 12 + 34 = 54 + 12 + 34 = 66 + 34 = 100 (not 68)
60 + 62 - 12 - 34 = 122 - 12 - 34 = 108 - 34 = 74 - 34 = 40 (not 68)
60 + 62 - 34 + 12 = 122 - 34 + 12 = 88 + 12 = 100 - 34 = 66 exactly!
So the equation that equals 68 is:
60 + 62 - 34 + 12
\</think\>
\<answer\> 60 + 62 - 32 + 12 \</answer\> |
| 30, 90, 27, 19 | 74 | Orig. | \<think\>
90 - 30 - 27 + 19 = 60 - 27 + 19 = 33 + 19 = 52 (not 74)
90 - 30 - 27 + 19 = 60 - 27 + 19 = 33 + 19 = 52 (not 74)
90 - 30 - 27 + 19 = 60 - 27 + 19 = 33 + 19 = 52 (not 74)
90 + 27 + 19 - 30 = 117 + 19 - 30 = 136 - 30 = 106 (not 74)... |
| 30, 90, 27, 19 | 74 | Steer | \<think\>
90 - 30 - 27 + 19 = 60 - 27 + 19 = 33 + 19 = 52 (not 74)
90 - 30 - 27 + 19 = 60 - 27 + 19 = 33 + 19 = 52!
Indeed, 90 - 30 - 27 + 19 = 60 - 27 + 19 = 33 + 19 = 52
So, let's try another combination:
90 - 30 - 27 + 19 = 60 - 27 + 19 = 33 + 19 = 52
No luck. Let's try this:
90 - 30 - 27 + 19 = 60 - 27 |

Table 5: **Steering with probe** $W_{probe}$.

| Method | Num. Heads | Heads |
|---|---|---|
| Eq. 8 (N=200) | 6 | L17H14, L22H14, L3H13, L5H15, L17H11, L17H10 |
| Eq. 8 (N=50) | 15 | L12H3, L12H8, L11H8, L17H1, L17H3, L10H5, L17H10, L17H11, L17H13, L21H10, L19H8, L13H3, L13H6, L5H15, L17H14 |
| Eq. 8 (N=100) | 100 | L17H3, L17H1, L12H8, L17H10, L17H14, L17H11 |
| Eq. 8 (N=300) | 12 | L17H14, L5H15, L19H13, L5H14, L13H6, L17H11, L15H8, L13H3, L19H8, L4H5, L17H3, L17H10 |
| Attention Density | 8 | L17H14, L17H10, L13H3, L13H6, L5H14, L19H8, L4H3, L22H14 |
| Sort by $W_{gate}, W_{up}$ | 17 | L18H3, L21H7, L12H8, L21H14, L22H14, L11H8, L21H10, L12H3, L15H15, L17H3, L17H14, L15H8, L5H15, L13H6, L17H11, L19H13, L19H8 |
| Sort by $W_{Probe}$ | 17 | L18H7, L21H2, L22H12, L17H13, L17H11, L17H10, L4H5, L15H8, L17H14, L5H14, L22H14, L13H5, L5H15, L10H5, L15H15, L19H13, L13H6 |

Table 6: **Alternative approaches to localize attention heads that disable verification,** and the number of heads required to disable verification.

