# OpenReview forum: "The Geometry of Self-Verification in a Task-Specific Reasoning Model"
_ICLR.cc/2026/Conference — ICLR 2026 Conference Withdrawn Submission_

### Official Review · Reviewer_5vi8 · 2025-10-31

**Soundness:** 2
**Presentation:** 2
**Contribution:** 2
**Rating:** 4
**Confidence:** 4

**Summary:**

This paper explores how task-specific reasoning models verify their own outputs. The authors train a model on the CountDown task using the DeepSeek-R1 recipe and conduct both top-down and bottom-up analyses to reverse-engineer its self-verification mechanism. In the top-down analysis, they use linear probes to identify GLU output vectors associated with verification, while the bottom-up analysis locates “previous-token” attention heads responsible for this process. Integrating the two perspectives shows that disabling a small number of these heads interrupts self-verification, indicating that they are essential components of the verification circuit. Similar subspaces are observed in both the base and general reasoning models, suggesting partial transferability. Overall, this study enhances the interpretability of internal computations in reasoning models.

**Strengths:**

1. This paper investigates an important question of how reasoning models perform self-verification, offering empirical insights into the mechanisms allowing models to check their own outputs.

2. The combination of top-down and bottom-up analyses provides complementary perspectives that identify concrete components, such as GLU vectors and attention heads, involved in self-verification.

3. The experimental results, particularly the causal interventions on attention heads, offer valuable evidence for understanding the internal processes that enable reasoning models to assess their own correctness.

**Weaknesses:**

1. Redundant expressions and insufficient motivation are found in the Introduction, which weakens the presentation of originality.

2. The experimental evaluation is narrow, only focusing on a single task and model, which restricts generalizability.

3. The proposed verification circuit is not tested under more diverse or challenging conditions, limiting the strength of the conclusions.

4. Robustness analyses are incomplete and lack comparisons between linear and nonlinear probes.

**Questions:**

1. The novelty of the paper is limited. It combines top-down and bottom-up analyses, but the latter has already been investigated in previous studies. Moreover, the paper does not sufficiently discuss the broader implications of the identified mechanism, reducing the overall impact of the contribution.

2. The Abstract, Introduction, and Discussion sections contain overlapping and redundant statements. In particular, the Introduction lacks a clear motivation and a distinct comparison with existing studies, making it difficult to convey the paper’s originality effectively.

3. The experimental evaluation focuses only on a single CountDown task with a relatively small dataset and one large language model. Such limited settings are inadequate to fully support the conclusions. The authors should broaden the evaluation to other mathematical reasoning or structurally similar tasks and include experiments on other open-source models.

4. The study should further test the identified verification circuit under different experimental conditions, e.g., scenarios with missing or noisy verification cues, to assess whether the mechanism remains effective in less favorable situations.

5. The paper also requires a more comprehensive robustness analysis, incorporating varied sampling strategies, layer-wise controls, and cross-seed experiments. The authors should compare linear and nonlinear probes and examine whether the “valid/invalid” GLU directions remain predictive after controlling for lexical and frequency-related effects.

**Details Of Ethics Concerns:**

No ethics concerns are identified in the paper.

---

### Official Review · Reviewer_NWGy · 2025-11-01

**Soundness:** 2
**Presentation:** 2
**Contribution:** 2
**Rating:** 2
**Confidence:** 4

**Summary:**

The paper studies how a model checks its own answers on the CountDown task (trained with an R1-style recipe). Because preference training makes the CoT very regular, the authors can compare hidden states at fixed steps. Top-down, simple probes and GLU vectors track “solved vs unsolved”; bottom-up, a few attention heads that look back at the target number are key—removing about six of them breaks self-verification and lowers those GLU activations. Similar patterns appear in the base model and a larger R1 model (via probe transfer). Limits: the head–GLU link is heuristic, interventions/parsing may confound, and results show necessity more than sufficiency, so it’s unclear how well this generalizes to tasks without explicit targets.

**Strengths:**

Tractable, interpretable setup: Leverages CountDown and preference-induced mode collapse to yield highly structured CoT, enabling precise time-step comparisons and clean causal interventions.

Middle-out causal localization: Combines top-down probes/GLU vectors with bottom-up previous-token heads via a heuristic inspired by inter-layer communication channels, isolating as few as six heads that reliably disable self-verification.

Geometric and cross-model evidence: Identifies GLUOut directions (and antipodes) aligned with success/failure semantics, accounts for SiLU effects, and replicates analogous components in both the base model and a larger R1 model (using EMB2EMB to transfer probes).

**Weaknesses:**

Limited generalization beyond context-anchored verification: The mechanism relies on explicit in-context targets; it may not capture “prior-based” verification, and APrev may reflect induction/copy behavior rather than genuine equality checking.

Methodological approximations and probe risks: The head–GLU scoring ignores QK routing, actual attention distributions, layer norms, and head interactions; linear probes may latch onto formatting/positional artifacts; strong sufficiency/rescue tests are missing.

Intervention and evaluation confounds: Zeroing WO can induce broader distribution shifts (e.g., control-flow/stopping disruptions leading to loops); parsing of “partial success/out-of-range” is brittle; evaluation uses relatively small sample sizes (300) with limited statistical reporting and threshold sensitivity analyses.  English–Chinese token asymmetry likely reflects tokenizer/data biases; the SiLU/antipode causal story is not directly validated; probe transfer (EMB2EMB) may misalign directions in the 14B model and confound “partial success” interpretations.

Necessity without demonstrated sufficiency: Ablating a small set of heads disables verification (necessity), but the paper does not show that these components are sufficient to re-enable or induce verification when other paths are suppressed (no rescue or overexpression tests).

Late definitions of “top-down” and “bottom-up”: These terms are only concretely instantiated in Section 4; earlier sections use them without precise operational definitions, creating avoidable confusion and cognitive load for readers mapping claims to methods.

**Questions:**

Please refer to weakness.

---

### Official Review · Reviewer_hhcZ · 2025-11-01

**Soundness:** 3
**Presentation:** 3
**Contribution:** 3
**Rating:** 6
**Confidence:** 4

**Summary:**

This paper uses MI techniques to understand how reasoning models do self-verification. From top-down, GLU helps while from bottom-up previous token heads are useful. From these insights, it is able to locate as few as six heads that can disable self-verification.

**Strengths:**

Overall, this is a good MI paper. The results are convincing and work across several reasoning models. The writing and presentation are clear, as it is easy to read and understand.

**Weaknesses:**

I have two comments:
1. In general, it is unclear how these findings can be applied in practice. The experiments show that attention heads can be identified to disable self-verification, but this is demonstrated only in a limited and toy setting. It would strengthen the paper to provide evidence that the approach generalizes to multiple reasoning tasks, which would make the work more impactful.
2. Section 5 should be integrated into the earlier sections. As it stands, it reads like a response to reviewers, which makes the overall structure of the paper scattered.

**Questions:**

see weakness

---

### Official Review · Reviewer_jRwt · 2025-11-03

**Soundness:** 3
**Presentation:** 3
**Contribution:** 2
**Rating:** 2
**Confidence:** 2

**Summary:**

The work studies the countdown task from a mechanistic interpretability perspective. After training a model until mode collapse, the authors study previous-token heads and find their role in ascertaining reasoning look into some special tokens that signify success or failure of subparts of the reasoning process. The study is conducted on Qwen2.5-3B and DeepSeek-R1(the latter with ICL).

**Strengths:**

Writing:
- The writing is good and can be followed well.

Contribution:
- The identification of circuits for self-verification is done correctly.
- The subspace idea for when self-verification occurs is a nice insight.

Code:
- Coda and datasets are provided.

**Weaknesses:**

Contribution:
- While the study highlights a few valid circuits, its contribution is limited. No mechanistic study of how the reasoning takes place is proposed, instead a few components (attention heads and special tokens) are identified that are necessary for the CoT to work. No general understanding of how the model solves the task at hand or into more general reasoning mechanisms is put forth. Given that the study is conducted on a simple synthetic task, a more complete understanding would be necessary.
- No methodological contribution: Standard tools from mechanistic interpretability are employed in a standard way.
- I think one would have to either provide a more complete mechanistic interpretability study for the countdown task or study the identified components across a wider range of tasks to crystallize their role in LLM reasoning.

**Questions:**

- I would be interested in a more thorough investigation for the countdown task to get a fuller mechanistic interpretability interpretation. What is the issue with that?
  - Do you think countdown is already too complex to provide that?
  - Do you have any identified any other (sub-)circuits?
- The other idea would be to stick with the limited set of identified (sub-)circuits:
  - Is it a general property that success/failure tokens and self-verification subspaces exist for other reasoning tasks?

**Details Of Ethics Concerns:**

-

---

### Note · Authors · 2025-11-28

I have read and agree with the venue's withdrawal policy on behalf of myself and my co-authors.